# Population genetic diversity and structure of the endangered species *Tetracentron sinense* Oliver (Tetracentraceae) with SNPs based on RAD sequencing

Zhong–Qiong Tian[1,2], Chao–Yang Jiang[2], Yu–Min Shu[2], Huan Zhang[3], Qiong La[1], Xiao-Hong Gan [2,3]*

**1** Key laboratory of southwest China wildlife resources conservation (Ministry of Education), China West Normal University, Nanchong, China, **2** Key Laboratory of Biodiversity and Environment on the Qinghai-Tibetan Plateau, Ministry of Education, Lhasa, China, **3** Institute of plant adaptation and utilization in southwest mountain, China West Normal University, Nanchong, China

* bhgan@cwnu.edu.cn

## Abstract

*Tetracentron sinense* Oliv. (*T. sinense*), as a tertiary living fossil, has experienced a significant decline in population numbers. Currently, genetic resources depletion and human activities have led to habitat fragmentation of relict and endangered plants, despite the abundant evidence of its medicinal, economic, and ecological value. Conservation strategies were clarified and evaluated based on the genetic structure characteristics and diversity patterns among 25 wild populations using Restriction site-associated DNA sequencing (RAD-seq) technology. Through SNP calling, filtering, genetic diversity analysis, discriminant analysis of principal components (DAPC), maximum-likelihood phylogenetic tree, and ADMIXTURE clustering, significant population structure and differentiation were identified. The results revealed a total of 2,169 single nucleotide polymorphisms (SNPs), indicating lower genetic variation but higher genetic differentiation (*He*: 0.10, *I*: 0.16, *Fst*: 0.33). Analysis of molecular variance (AMOVA) showed that genetic variation within populations accounted for 77% of the total variance. DAPC, maximum-likelihood phylogenetic tree, and ADMIXTURE clustering analysis grouped the 25 populations into five distinct clades influenced by isolation, restricted gene flow, and complex topography. To preserve the genetic integrity of *T. sinense*, it is recommended to establish conservation units corresponding to different geographic clades, with a focus on populations with low/high genetic diversity by implementing artificial reproduction and germplasm resource nurseries. Given the species' vulnerable conservation status, urgent implementation of the aforementioned conservation strategies is necessary to safeguard the remaining genetic resources.

**Data availability statement:** The data in our study is deposited on Figshare website (https://doi.org/10.6084/m9.figshare.22209688.v1).

**Funding:** XHG NO.32070371 and No.2018PC001 supported by National Natural Science Foundation of China. Xiaohong Gan: Funding acquisition; Investigation; Project administration; Supervision; Visualization; Writing–review & editing.

**Competing interests:** The authors have declared that no competing interests exist.

**Abbreviations: RAD-seq**, Restriction site–associated DNA sequencing; **SNPs**, Single Nucleotide Polymorphisms; **DAPC**, Discriminant analysis of principal components; **PCA**, Principal Component Analysis; **AMOVA**, Analysis of Molecular Variance; **ESUs**, Evolutionary Significant Units; **NGS**: Next Generation Sequencing; **CTAB**, Hexadecyl trimethyl ammonium Bromide; **PCR**, Polymerase Chain Reaction; **CV**, Cross Validation Error.

## Introduction

Genetic diversity represents a cornerstone of biodiversity conservation, with its preservation recognized as critical for maintaining ecosystem resilience and evolutionary potential [1]. Conservation strategies prioritize the maintenance of genetic variation both within and among populations, as this variation underpins species' capacity to adapt to environmental change. Declines in genetic diversity heighten inbreeding risks and erode adaptive potential, thereby threatening the long-term viability of populations [2]. The erosion of genetic diversity in endangered plant species significantly elevates extinction vulnerability by reducing adaptive capacity and ecological resilience. Consequently, comprehensive characterization of genetic architecture—including population structure, differentiation patterns, and allelic variation—constitutes an imperative precursor to effective conservation planning. Such analyses enable evidence-based prioritization of conservation units and management interventions, particularly for taxa like *T. sinense* that persist in fragmented populations with critically low effective population sizes [3,4].

As the sole extant species within the genus Tetracentron, *T. sinense* is endemic to East Asian riparian ecosystems and forest understories [5]. This Arcto-Tertiary relict, often termed a 'living fossil' due to its morphological conservatism, exhibits a robust paleobotanical record traceable to the Eocene epoch (53.0–36.5 million years ago) [6]. *T. sinense* holds paramount importance for its economic applications (furniture, ornamental horticulture, timber production) and medicinal properties (e.g., anticancer, antibacterial compounds). However, unsustainable harvesting, anthropogenic disturbance, and habitat fragmentation have precipitated a marked reduction in population size [7]. Consequentially, this species is listed in Appendix III of CITES [8,9]. Despite its conservation priority, population genetic studies leveraging high-throughput sequencing (HTS) technologies remain scarce for *T. sinense*.

Previous investigations on *T. sinense* have addressed diverse ecological and physiological aspects, including pollination biology, seedling establishment, community dynamics, photosynthetic ecophysiology, and population structure [4,10–17]. However, comprehensive population genetic studies remain conspicuously limited. Analyses using chloroplast sequences and inter-simple sequence repeat (ISSR) markers have identified distinct genetic clusters among populations [4,18,19]. Moreover, genetic variation assessments revealed the presence of evolutionarily significant units (ESUs) across its range. Critical threats to its survival—such as inefficient pollination systems, reduced genetic diversity, and geographic isolation—have been identified as primary drivers of endangerment. The recent completion of a high-quality chromosome-level genome assembly for *T. sinense* has underscored the critical need for developing robust genetic markers to comprehensively evaluate population genetic diversity and adaptive potential [19].

Population genetic studies in widely distributed species provide a robust statistical framework for understanding natural population evolution. Advances in HTS, or next-generation sequencing (NGS), have enabled the genomic development of numerous genetic markers. Restriction site-associated DNA sequencing (RAD-seq) using Illumina platforms is cost-effective and produces highly reliable data, making it

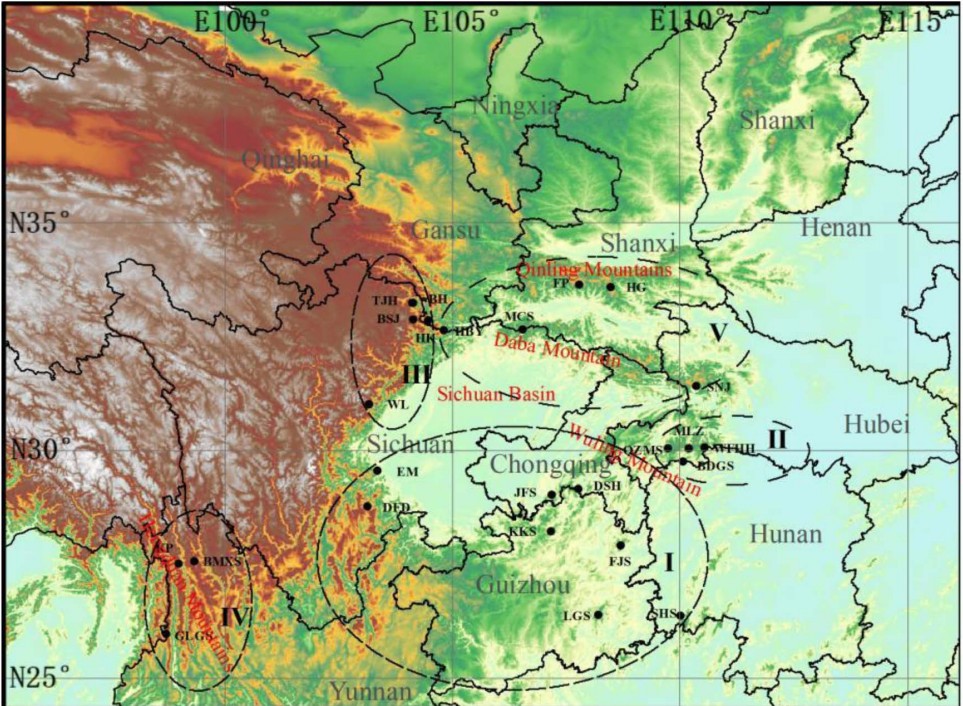

**Fig 1. Geographic distribution of 25 *T. sinense* populations.** Notes: reprinted from Li, S., Gan, X., Han, H. et al. Low within-population genetic diversity and high genetic differentiation among populations of the endangered plant T. sinense revealed by inter-simple sequence repeat analysis. Annals of Forest Science 75, 74 (2018). under a CC BY license, with permission from Annals of Forest Science, original copyright 2018.

a powerful tool for population genomic analyses [20]. Single nucleotide polymorphisms (SNPs) represent the most abundant form of genomic variation within species, providing detailed insights into population genetic architecture [21]. SNP genotyping has emerged as a powerful tool for characterizing natural population evolution and informing conservation genetics. In this study, we utilized SNPs to characterize the genetic diversity, structure, and differentiation of *T. sinense* [22]. By leveraging a large SNP dataset, we investigated key population genetic parameters critical for understanding evolutionary processes. The methodological framework and findings presented here are anticipated to facilitate the development and implementation of targeted conservation strategies for this endangered species.

## Materials and methods

### Sample collection and DNA extraction

*T. sinense* is distributed primarily in central and southern China. A total of 164 individuals were collected from 25 wild populations across China (Fig 1, Table 1, S1 Table). Samples were collected with a minimum spatial distance of 50 meters to minimize kinship bias, and leaf tissues were preserved using silica gel desiccation. Genomic DNA was extracted from dried leaves via the cetyltrimethylammonium bromide (CTAB) protocol [23]. Voucher specimens were deposited in the herbarium of China West Normal University (Specimen ID: MCS202005).

### RAD library preparation and sequencing

RAD-seq libraries were constructed following the protocol described by Baird et al. [24], with minor modifications. Genomic DNA was first digested with the restriction endonuclease TaqαI. Digested fragments were then ligated to P1

**Table 1. The geographical location, population code, and sample size of 25 *T. sinense* populations.**

| Code | Population | Number | Longitude (E) | Latitude (N) | Altitude (m) |
|------|------------|--------|---------------|--------------|--------------|
| P1 | JFS | 5 | 107.18 | 29.02 | 2077—2125 |
| P2 | HG | 11 | 108.48 | 33.58 | 1667—1710 |
| P3 | FP | 5 | 107.78 | 33.65 | 1602—1658 |
| P4 | BDGS | 10 | 110.06 | 29.76 | 1209—1502 |
| P5 | MLZ | 6 | 110.2 | 30.04 | 1435—1608 |
| P6 | TJH | 10 | 104.82 | 32.64 | 1943—2041 |
| P7 | HK | 7 | 104.37 | 32.9 | 1720—1782 |
| P8 | WL | 9 | 103.17 | 31.01 | 2400—2430 |
| P9 | KP | 4 | 99.13 | 27.55 | 2561—2583 |
| P10 | BMXS | 8 | 99.35 | 27.63 | 2562—2902 |
| P11 | QZMS | 5 | 109.74 | 30.04 | 1444—1552 |
| P12 | LGS | 6 | 108.2 | 26.38 | 1644—1976 |
| P13 | MCS | 5 | 106.55 | 32.65 | 1873—1934 |
| P14 | FJS | 4 | 108.69 | 27.91 | 2079—2103 |
| P15 | BH | 7 | 104.14 | 33.25 | 1867—1945 |
| P16 | DFD | 6 | 103.14 | 28.77 | 2163—2318 |
| P17 | SHS | 7 | 110.01 | 26.37 | 1492—1704 |
| P18 | SNJ | 5 | 110.36 | 31.41 | 1256—1737 |
| P19 | HBY | 8 | 104.12 | 33.24 | 2037—2060 |
| P20 | KKS | 5 | 107.17 | 28.23 | 1614—1626 |
| P21 | DSH | 3 | 107.76 | 29.15 | 1627—1637 |
| P22 | BSJ | 10 | 104.33 | 32.9 | 2182—2302 |
| P23 | EM | 4 | 103.36 | 29.56 | 1616—1847 |
| P24 | WFHH | 9 | 110.54 | 30.07 | 935—1449 |
| P25 | GLGS | 5 | 98.71 | 25.97 | 2472—2499 |

adapters containing sequences for PCR amplification, Illumina sequencing, and sample-specific barcodes, along with the TaqαI recognition site. After adapter ligation, DNA fragments were size-selected via agarose gel electrophoresis to isolate target bands in the 300–500 bp range. The selected fragments were ligated to P2 adapters, and paired-end libraries with 150 bp insert sizes were prepared for sequencing on the Illumina HiSeqTM platform at Majorbio Company (Shanghai, China).

## Sequence alignment and SNP genotyping

Raw sequencing data were demultiplexed using barcodes to assign reads to individual samples. Adapter-trimmed reads were evaluated for sequencing quality and data volume. To ensure transparency, we explicitly declare that we performed de novo assembly of the genome using Stacks v2.4 software [25], a critical step for readers unfamiliar with the workflow. The Stacks v2.4 [25] (ulian Catchen and Nicolas Rochette, Urbana, IL, USA) software was utilized for clustering sequence fragments post quality control, followed by SNP detection and filtering using the resulting vcf-file with VCFtools v.0.1.16 [26]. Initial processing included: (1) adapter removal and 5' -end trimming of non-AGCT bases; (2) quality trimming of low-quality ends (quality score < 20); (3) removal of reads with ≥10% ambiguous bases (Ns); and (4) exclusion of fragments shorter than 25 bp after trimming. Clean reads were mapped to samples to ensure data integrity for downstream analysis.

### Genetic diversity

Genetic diversity parameters for *T. sinense* were estimated using Stacks v2.4 [25], including observed allele count (*Na*), effective allele count (*Ne*), expected heterozygosity (*He*), Shannon information index (*I*), and nucleotide diversity (π). Neutrality tests were conducted using VCFtools v0.1.16 [26] to compute *Tajima' s D* at 95% confidence intervals across all loci [27,28]. Analysis of variance (ANOVA) and post-hoc multiple comparisons were performed to evaluate diversity index differences among the 25 populations [29]. Molecular variance partitioning was implemented in Arlequin v3.5 to quantify significant genetic variation among and within populations [30].

### Population structure, phylogenetic tree, and principal components analysis

To characterize the genetic structure of the 25 *T. sinense* populations, SNPs were first filtered using PLINK v1.9 [31], followed by genetic structure estimation with ADMIXTURE v1.3.0 [32]. Genetic cluster numbers (K) ranging from 1 to 27 were evaluated via 10-fold cross-validation (CV), with 27 independent runs performed for each K value. The optimal K was determined based on the lowest CV error in the ADMIXTURE analysis. Additionally, the optimal population structure of 164 *T. sinense* individuals from 25 populations was inferred using adegenet v2.1.11 [33], integrated with geographic coordinates. A maximum-likelihood (ML) phylogeny was reconstructed in RAxML v7.3 [34] under the GTRGAMMA substitution model with 1000 bootstrap replicates. Population phylogenetic trees were generated using the RAxML ML algorithm. Individual clustering patterns were visualized via principal component analysis (PCA) based on SNP-derived genetic distances. Genetic relationships among individuals were characterized using discriminant analysis of principal components (DAPC) implemented in the adegenet package (R v3.6.1) [33]. Kinship coefficients were estimated with GCTA v1.92.1 [35], while identity-by-state (IBS) genetic distances were computed using PLINK v1.90 [31] to generate a distance matrix.

### Population linkage disequilibrium analyses

Pairwise linkage disequilibrium (LD) among 2,169 genome-wide SNPs across all scaffolds in 164 Tetracentron sinense individuals was estimated using allele frequency correlations (r²) via PopLDdecay v3.41 [36]. LD analysis quantified r² values to characterize marker associations. Population genetic differentiation (*FST*) was calculated using VCFtools v0.1.16 [26], while gene flow (*Nm*) estimates were derived from *Fst* values using the equation:

$$Nm = 0.25 * \frac{1 - Fst}{Fst}$$

### Ethics statement

Because this study was carried out on an endangered species, I confirmed that I complied with all relevant institutional, national and international guidelines. This study was supported by National Natural Science Foundation of China, including handling these plants and collecting samples. The Nature Reserve Authority granted permissions to Professor Xiao–hong Gan for using the endangered species of plant (*T. sinense*).

## Results

### Quality and processing of sequence data

A total of 164 samples from 25 *T. sinense* populations (JFS, HG, FP, BDGS, MLZ, TJH, HK, WL, KP, BMXS, QZMS, LGS, MCS, FJS, BH, DFD, SHS, SNJ, HBY, KKS, DSH, BSJ, EM, WFHH, GLGS) were subjected to RAD-seq for SNP identification. Sequencing with the Taqαl restriction enzyme yielded 1,755 million base pairs (Mbp) of raw data. Clean reads

per sample ranged from 6,917,919–26,051,926, with sequence quality scores (Q30) averaging 94.62% (range: 87.37–96.27%). Genome-wide GC content varied from 41.22% to 59.54%, with a mean of 47.24% (S2 Table, S1 Fig). Following initial quality filtering, 6,158,926 SNPs were identified. After applying stringent filtering criteria (missing rate < 50%, minor allele frequency ≥ 0.05), 38,097 high-quality SNPs were retained for genotyping analysis. Subsequent population genetic diversity and structure analyses utilized 2,169 polymorphic markers derived from high-quality unlinked SNPs.

## Genetic diversity at species and population levels

We summarized the genetic diversity parameters of *T. sinense* at the species-level. The average values of *Na*, *Ne*, *He*, *I*, and π were 2.0, 1.17, 0.13, 0.24, and 0.23, respectively (Tables 2, 3, and S3). Based on the entire dataset analyzed using VCFtools [26], the *Tajima' s D* value at the species-level was 0.67 (Table 3). Notably, the significantly positive *Tajima' s D* values suggest a potential decrease in population size and/or the action of balancing selection [27].

Genetic diversity parameters for *T. sinense* are summarized in Tables 2, 3, and S2. At the species level, average values for observed alleles (*Na* 2.00), effective alleles (*Ne* = 1.17), expected heterozygosity (*He* = 0.13), Shannon' s index (*I* = 0.24), and nucleotide diversity (π = 0.23) were recorded. *Tajima' s D* analysis of the full dataset using VCFtools [26,27] yielded a species-level value of 0.6741 (Table 3). Notably, the significantly positive *Tajima' s D* suggests potential population size reduction and/or balancing selection acting on the species [27].

**Table 2. Genomic diversity statistics calculated at the species and population level of 25 *T. sinense* populations.**

| Population | Na | Ne | He | I |
|---|---|---|---|---|
| JFS | 1.4325 ± 0.4918 | 1.1605 ± 0.2570 | 0.1046 ± 0.1481 | 0.1690 ± 0.2213 |
| HG | 1.3151 ± 0.4567 | 1.1284 ± 0.2374 | 0.0834 ± 0.1393 | 0.1333 ± 0.2105 |
| FP | 1.3630 ± 0.4802 | 1.1907 ± 0.3128 | 0.1148 ± 0.1735 | 0.1755 ± 0.2532 |
| BDGS | 1.4985 ± 0.4987 | 1.1730 ± 0.2557 | 0.1152 ± 0.1476 | 0.1887 ± 0.2212 |
| MLZ | 1.3866 ± 0.4869 | 1.1734 ± 0.2773 | 0.1106 ± 0.1594 | 0.1742 ± 0.2379 |
| TJH | 1.2788 ± 0.4468 | 1.1606 ± 0.2855 | 0.0984 ± 0.1637 | 0.1490 ± 0.2434 |
| HK | 1.3791 ± 0.4768 | 1.1644 ± 0.2709 | 0.1045 ± 0.1552 | 0.1650 ± 0.2313 |
| WL | 1.2866 ± 0.4522 | 1.1442 ± 0.2655 | 0.0908 ± 0.1537 | 0.1406 ± 0.2307 |
| KP | 1.3287 ± 0.4695 | 1.1538 ± 0.2689 | 0.0974 ± 0.1544 | 0.1526 ± 0.2314 |
| BMXS | 1.2786 ± 0.4478 | 1.1512 ± 0.2860 | 0.0917 ± 0.1607 | 0.1399 ± 0.2368 |
| QZMS | 1.3827 ± 0.4845 | 1.1664 ± 0.2798 | 0.1051 ± 0.1585 | 0.1658 ± 0.2347 |
| LGS | 1.4567 ± 0.4977 | 0.9937 ± 0.2512 | 0.1067 ± 0.1472 | 0.1742 ± 0.2213 |
| MCS | 1.3582 ± 0.4892 | 0.9984 ± 0.2640 | 0.1075 ± 0.1523 | 0.1722 ± 0.2286 |
| WL | 1.3778 ± 0.4931 | 1.1805 ± 0.2591 | 0.1137 ± 0.1502 | 0.1774 ± 0.2252 |
| BH | 1.3471 ± 0.4905 | 1.1546 ± 0.2586 | 0.0999 ± 0.1495 | 0.1578 ± 0.2241 |
| DFD | 1.2838 ± 0.4998 | 1.1659 ± 0.2540 | 0.0992 ± 0.1466 | 0.1497 ± 0.2197 |
| SHS | 1.3928 ± 0.4604 | 1.1844 ± 0.2557 | 0.1158 ± 0.1492 | 0.1809 ± 0.2250 |
| SNJ | 1.3684 ± 0.4881 | 1.1665 ± 0.2855 | 0.1070 ± 0.1617 | 0.1686 ± 0.2397 |
| HBY | 1.3012 ± 0.4864 | 1.1318 ± 0.2626 | 0.0840 ± 0.1514 | 0.1329 ± 0.2267 |
| KKS | 1.4042 ± 0.4577 | 1.1605 ± 0.2531 | 0.1041 ± 0.1454 | 0.1669 ± 0.2181 |
| DSH | 1.3943 ± 0.4811 | 1.1779 ± 0.2704 | 0.1122 ± 0.1564 | 0.1762 ± 0.2350 |
| BSJ | 1.3051 ± 0.4881 | 1.1410 ± 0.2904 | 0.0900 ± 0.1645 | 0.1413 ± 0.2439 |
| EM | 1.4882 ± 0.4502 | 1.1660 ± 0.3021 | 0.1108 ± 0.1682 | 0.1820 ± 0.2468 |
| WFHH | 1.4062 ± 0.4758 | 1.1563 ± 0.2601 | 0.1018 ± 0.1522 | 0.1639 ± 0.2298 |
| GLGS | 1.4269 ± 0.4838 | 1.1614 ± 0.2856 | 0.1060 ± 0.1627 | 0.1711 ± 0.2424 |
| Species | 1.3696 ± 0.4918 | 1.1482 ± 0.2570 | 0.1029 ± 0.1481 | 0.1627 ± 0.2213 |

**Table 3. *Pi* and *Tajima' s D* values among 5 clades of *T. sinense*.**

| Clades | Nucleotide diversity value (*Pi*) | Tajima' s D |
|---|---|---|
| I | 414.61 | 0.2406 |
| II | 349.82 | 0.2837 |
| III | 415.7 | 0.6903 |
| IV | 316.5 | 1.1195 |
| V | 370.23 | 0.4011 |

At the population level, genetic diversity parameters varied as follows: *Na* ranged from 1.28 (GLGS) to 1.50 (BSJ), *Ne* from 1.00 (HG) to 1.19 (BMXS), *He* from 0.08 (MLZ) to 0.12 (BSJ), and *I* from 0.13 (MLZ) to 0.19 (BSJ) (Table 2). Congruent trends were observed in *He* and *I* metrics. Notably, WFHH, BDGS, and MLZ populations displayed relatively high genetic diversity, whereas HBY and HG populations exhibited significantly lower values (Table 2).

## Population genetic structure, principal components analysis, and phylogenetic analysis

To characterize population genetic relationships, genetic structure analysis was performed using adegenet and ADMIXTURE v1.3.0 [32,33]. Adegenet analysis revealed K = 5 as the optimal clustering solution for the 164 *T. sinense* individuals, with the sum of squares reaching a plateau at this value (Fig 2A). Principal component analysis (PCA) results, congruent with adegenet findings, also supported five genetic clusters (Fig 2B). Cluster composition was as follows, Cluster1: BH, BMXS, BSJ, GLGS, HK, KP, TJH, WL; Cluster2: DFD, FJS, JFS, LGS, SHS; Cluster3: DSH, KKS; Cluster4: BDGS, FP, HBY, HG, MCS, QZMS, SNJ, WFHH, MLZ; Cluster5: EM (monotypic). Detailed cluster assignments are provided in Supplementary S4 Table.

The optimal number of genetic clusters (K) for the 164 *T. sinense* individuals was determined via CV error minimization in ADMIXTURE analysis. CV errors ranged from 0.35 to 0.66 across K values 1–27, with the lowest error (0.35) observed at K = 5 (Fig 2C, Supplementary S5 Table). Complete results for all tested K values (1–27) are presented in Supplementary S2 Fig. At K = 5, individuals from 25 populations were partitioned into five genetically distinct clades with maximal probability: CladeI: 8 populations from the Hengduan and Wuling Mountains; CladeII: 4 populations from the Wuling Mountains; CladeIII: 5 populations from the Minshan Mountains; CladeIV: 3 populations from the Hengduan Mountains; CladeV: 5 populations from the Qinling-Daba Mountains. Geographic distributions and clade assignments are visualized in Fig 2D and detailed in Supplementary S6 Table.

DAPC was performed to characterize genetic structure among 164 *T. sinense* individuals. Principal components 1–3 (PC1–PC3) accounted for 33.62% of cumulative variance (Fig 3A and B, Supplementary S7 Table). DAPC plots revealed distinct clustering of the five genetic groups identified in ADMIXTURE analysis, confirming their genetic divergence.

A ML phylogenetic tree was reconstructed using high-quality unlinked SNPs with RAxML v7.3 [34]. The tree exhibited strong bootstrap support across most branches, validating its reliability. Five major clades were resolved, CladeI: DFD, EM, SHS, LGS, KKS, JFS, DSH, FJS; CladeII: SNJ, BDGS, MLZ, QZMS, WFHH; CladeIII: TJH, BSJ, BH, HK, WL; CladeIV: KP, BMXS, GLGS; CladeV: SNJ, MCS, HG, HBY, FP. Subclustering within CladeI and CladeIII was observed; CladeI-Subclade1: KKS, JFS, FJS, LGS, SHS; CladeI-Subclade2: DFD, EM, DSH; CladeIII-Subclade1: BSJ, BH, TJH; CladeIII-subclade2: WL, HK, BH. These results are visualized in Fig 3C.

## Population differentiation, LDdecay, and AMOVA analyses

Population differentiation coefficients (*Fst*) among the 25 *T. sinense* populations averaged 0.33 (range: 0.14–0.50) across 164 individuals (Table 4). According to *Wright' s* guidelines, *Fst* values >0.25 indicate substantial population divergence. In this study, *Fst* values between certain clades fell below 0.25, whereas others exceeded this threshold, with the highest

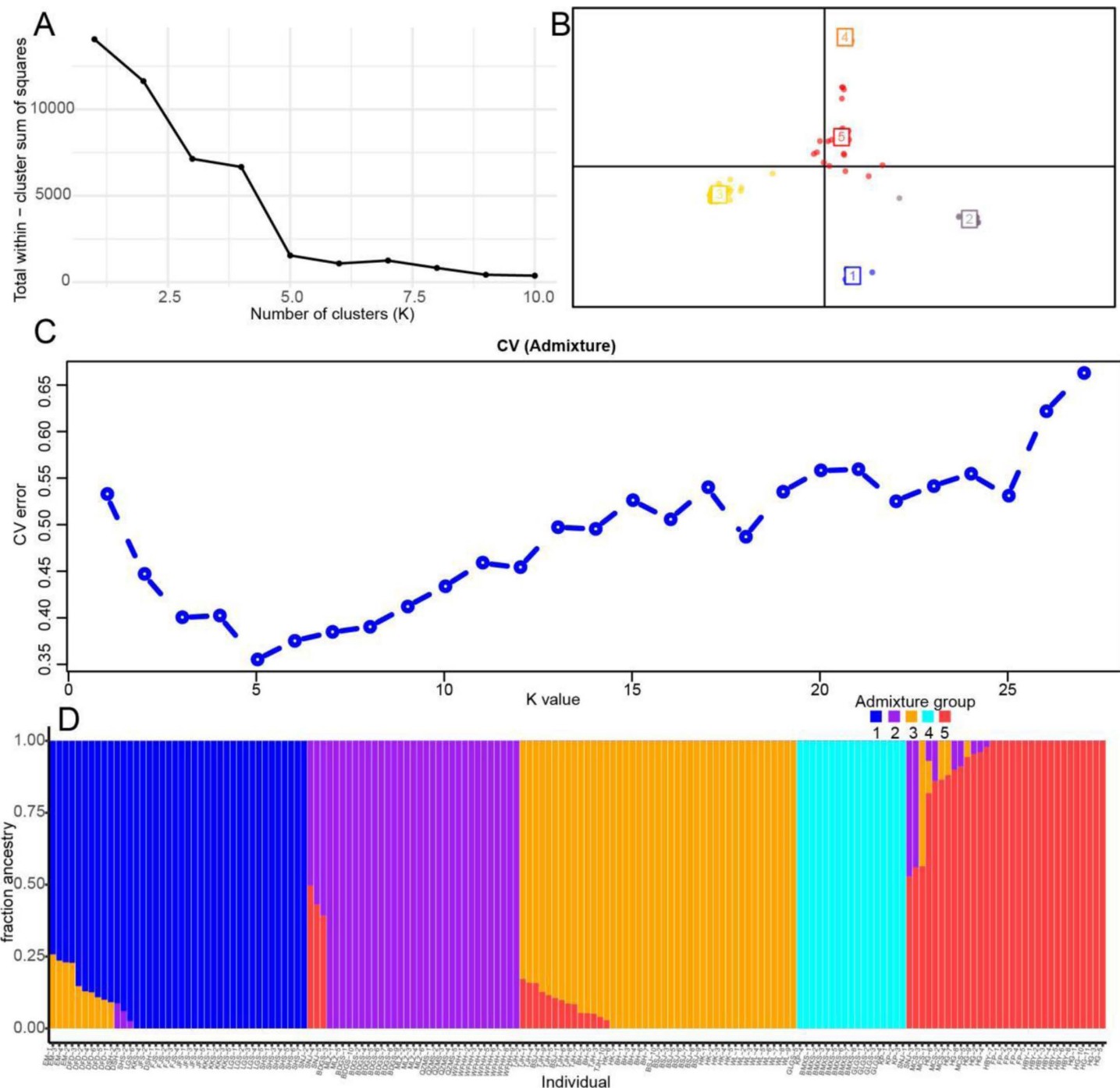

**Fig 2. Population genetic structure of *T. sinense* according based on the adegenet and ADMIXTURE analysis.** A, Best choice of number of clusters based on adegenet analysis; B, PCA result of the five clusters based on adegenet analysis; C, the patterns are shown with the CV error value for the K value (K = 2–27) showing the best K = 5; D, the five clades are represented in different colors (CladeI-Vare denoted by blue, purple, yellow, green and red, respectively).

differentiation observed between CladesII and IV ($Fst = 0.50$). Gene flow ($Nm$) estimates among the five clades ranged from 0.28 (CladesIV and V) to 1.53 (CladesII and V) (Table 5).

Analysis of molecular variance (AMOVA) indicated that 77% of total genetic variance resided within populations, with only 23% distributed among populations (Table 6). This pattern suggests that the majority of genetic diversity in *T. sinense* occurs within populations. A calculated $Nm$ value of 1.66 indicates moderate gene flow among populations.

LD dynamics were characterized across the five inferred clades. CladeI exhibited the slowest LD attenuation, with r² values remaining ≥0.94 up to 9 kb. Conversely, CladeV displayed the most rapid LD decay, where r² dropped below 1.00 at ~3 kb, followed by CladesIII and IV (r² = 1.00 at 9 kb). CladeII maintained r² ≥ 1.00 until ~15 kb. Integrating evidence from high nucleotide diversity, phylogenetic relationships, and the shortest LD decay distance, CladeV was hypothesized to represent the evolutionary origin center (Fig 4, Table 7). Geographically, CladeV and CladeIV were the most isolated ($Fst = 0.47$), indicating strong genetic differentiation. Additionally, CladeIV harbored the lowest genetic diversity among clades, further supporting the Qinling-Daba Mountains as the species' origin center.

## Discussion

Effective conservation of endangered species requires characterization of genetic diversity and population structure. *He* and *I* are widely recognized as key metrics of genetic diversity, where higher values reflect greater genetic variability and reduced population uniformity. In this study, interpopulation *He* and *I* averaged 0.10 and 0.16, respectively (Table 2). Compared to other endangered taxa—*Tetraena mongolica* (He = 0.35, I = 0.27), *Davidia involucrata* (He = 0.34, I = 0.51), *Chosenia arbutifolia* (He = 0.37, I = 0.53), and *Sinowilsonia fienryi* (He = 0.28, I = 0.40)—*T. sinense* exhibited significantly lower genetic diversity [37–40]. Populations from Sichuan and Guizhou provinces showed elevated diversity, likely due to Southwest China' s favorable climatic conditions (humidity and temperature) supporting species persistence. Conversely, GLGS, BMXS, and KP populations in Yunnan displayed reduced diversity, potentially resulting from thermal stress. Despite overall low genetic diversity, a moderate gene flow estimate ($Nm = 1.66$) indicates residual adaptive potential in wild *T. sinense* populations. Previous research highlights plant breeding systems and gene flow as critical determinants of genetic diversity [41,42]. Limited pollen dispersal and short-distance seed movement in *T. sinense* likely exacerbate inbreeding depression, increasing homozygosity and exposing deleterious recessive alleles. These factors contribute to the species' endangered status, marked by skewed age structures with abundant juveniles but few mature individuals.

Genetic differentiation in Tetracentron sinense arises from long-term genetic isolation and species-specific biological traits. Analysis of *Fst*, a metric quantifying clade-level divergence, revealed an average genetic differentiation of 0.33 among the five clades, indicating substantial population subdivision. AMOVA results further showed that 77% of genetic variation resided within populations. While conservation initiatives have prioritized protecting populations in nature reserves, the scarcity of mid-aged individuals poses challenges for population renewal. High intrapopulation genetic diversity suggests the species retains adaptive potential for environmental shifts.

Sampling sites for *T. sinense* are situated in montane nature reserves, preserving habitat heterogeneity. Significant genetic differentiation among populations has resulted from prolonged geographic isolation. Comparable trends of reduced genetic variation and elevated differentiation have been documented in other endangered taxa, including *Loropetalum subcordatum*, *Omphalogramma souliei*, and *Cypripedium japonicum* [43–45]. Climate change and anthropogenic activities have driven recent population declines in *T. sinense*. Collectively, these results suggest that reduced genetic variation and elevated differentiation in *T. sinense* stem from prolonged isolation, climate change, and anthropogenic pressures.

Consistent genetic clustering into five clades was inferred by adegenet and Admixture analyses for 164 *T. sinense* individuals across 25 populations. Subtle discrepancies in cluster composition—adegenet Cluster1 encompassing Admixture CladesIII and IV, adegenet Cluster4 merging Admixture CladesII and V, and adegenet Clusters2 and 3 corresponding to Admixture CladeI—likely arise from algorithmic differences between software platforms. Despite these nuances, the overall congruence in clustering patterns validates the robustness of the inferred population structure. Using RAD-seq, we

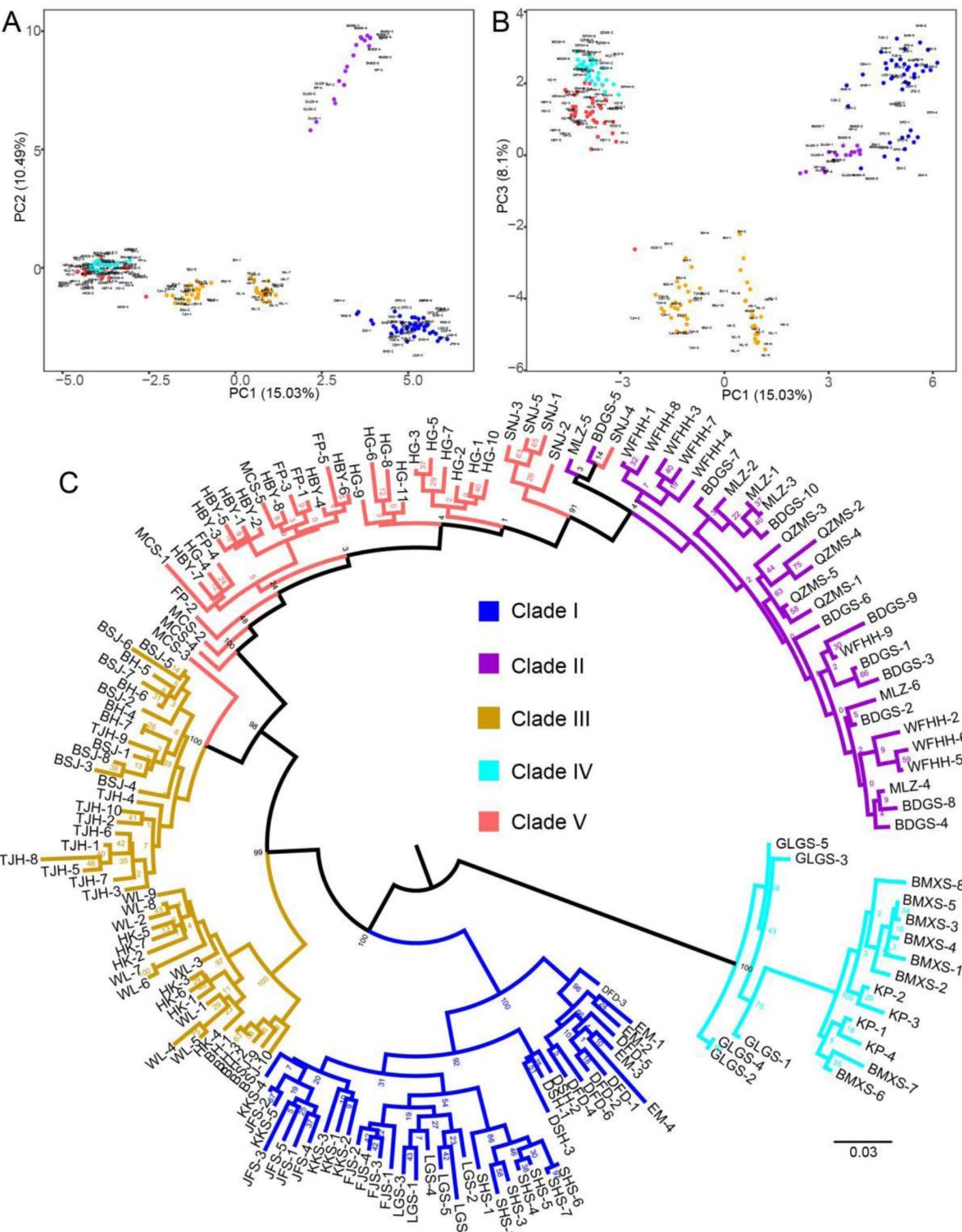

**Fig 3. DAPC and Maximum-likelihood (ML) phylogenetic analysis of *T. sinense* samples, with the proportion of the variance explained being 15.03% for PC1, 10.49% for PC2 and 8.1% for PC3.** A, PC1 V PC2; B, PC1 V PC3; C, ML phylogenetic tree of 164 individuals from 25 *T. sinense* populations constructed with the genome–wide SNP dataset.

**Table 4. *Fst* values among five clades of *T. sinense*.**

| Clades | II | III | IV | V |
|---|---|---|---|---|
| I | 0.3350 | 0.2406 | 0.4192 | 0.3072 |
| II | | 0.2837 | 0.4970 | 0.1407 |
| III | | | 0.4341 | 0.2092 |
| IV | | | | 0.4736 |

**Table 5. *Nm* analysis among five clades of *T. sinense*.**

| Clades | II | III | IV | V |
|---|---|---|---|---|
| I | 0.50 | 0.79 | 0.35 | 0.56 |
| II | | 0.63 | 0.25 | 1.53 |
| III | | | 0.33 | 0.95 |
| IV | | | | 0.28 |

**Table 6. Analysis of molecular variance (AMOVA) analysis of genetic variation within and among populations in *T. sinense*.**

| Source | df | SS | MS | Est.var. | Percent (%) | P |
|---|---|---|---|---|---|---|
| Among Pops | 23 | 12715.752 | 529.823 | 53.828 | 23 | |
| Within Pops | 77 | 24795.352 | 178.384 | 178.384 | 77 | |
| Total | 163 | 37511.104 | | 232.212 | 100 | 0.01 |
| Nm | 1.657 | | | | | |

Notes: *df*, degrees of freedom; *SS*, sum of squares deviation; *MS*, mean squared deviation; *Est. var.*, estimates of variance components; and %, percentage of total variance contributed by each component.

identified five geographically structured genetic clusters in *T. sinense*, aligning with Sun et al.' s [18] K = 5 optimal solution and previous refugia hypotheses [4,18,46]. Geographically, CladeI occupied the southern Sichuan Basin (DFD/EM/DSH on the Sichuan-Guizhou border and KKS/JFS/FJS/LGS/SHS near Wuling Mountains), CladeII spanned northern Wuling and southern Qinling-Daba Mountains, CladeIII resided in northern Minshan Mountains, CladeIV occupied southwestern Hengduan Mountains, and CladeV inhabited Qinling-Daba Mountains. These clades are partitioned by major mountain systems (Hengduan, Wuling, Minshan, Qinling-Daba), underscoring historical isolation' s role in shaping distributions [4]. Comparative analysis revealed: (1) Li et al.' s ClusterI corresponds to our merged CladesI and III, while their ClusterII aligns with our CladesII and V; (2) Sun et al. [18] classified Gansu populations (RW/RT) as a distinct CladeI, whereas we integrated them into broader phylogeographic groupings. Population genetic structure reflects spatiotemporal genetic variation distribution [41,47], with this study' s findings advancing understanding of *T. sinense*'s evolutionary history.

These findings indicate that genetic clustering in *T. sinense* is primarily structured by major mountain systems (Hengduan, Wuling, Minshan, and Qinling-Daba Mountains), which have historically served as refugia for multiple taxa in the region, including *Oxyria sinensis*, *Roscoea humeana*, *Pinus armandii*, and *Eurycorymbus cavaleriei* [48–51]. Geographical isolation mediated by these ranges likely contributes to the species' low genetic variation, high differentiation, and unique genetic structure. The observed disjunct distributions are hypothesized to result from intense Quaternary glaciations and tectonic activity (e.g., Tibetan Plateau uplift) [52–56]. For isolated populations, restricted long-distance gene flow—exacerbated by topographic barriers—may drive patterns similar to those documented in *Phellodendron amurense*, *Tetraena mongolica*, and *Gymnocarpos przewalskii* [39,57,58]. Prolonged geographic isolation can lead to reproductive isolation and elevated extinction risk, as reflected in the fragmented genetic structure of *T. sinense* populations.

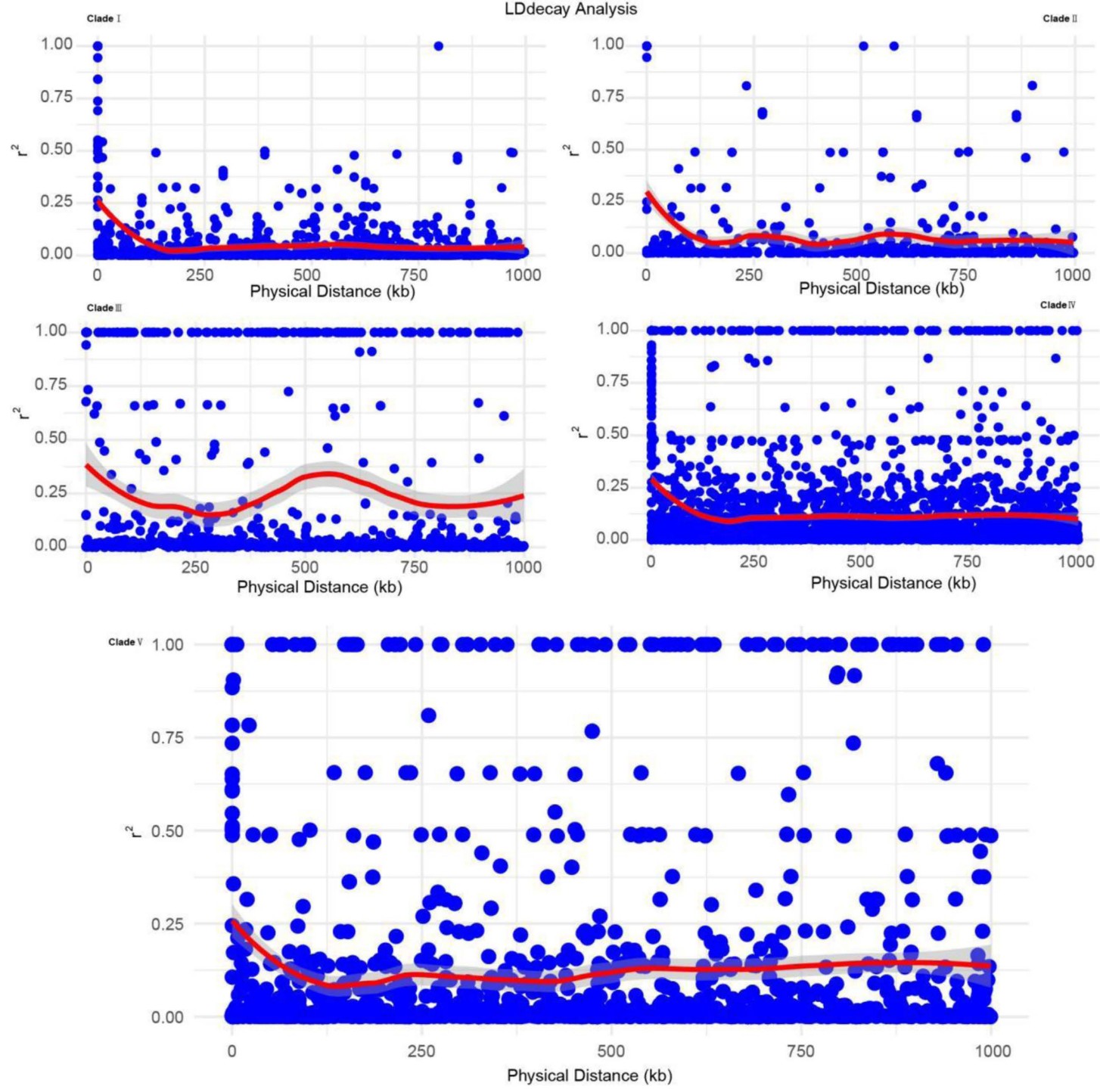

**Fig 4. LDdecay analysis of the five clades.**

**Table 7. Linkage disequilibrium decay statistics of r² and LD distance of five clades.**

| Clades | max LD r2 | half LD r2 | LD length |
|---|---|---|---|
| I | 0.94 | 0.4722 | 9 |
| II | 1.00 | 0.50 | 15 |
| III | 1.00 | 0.50 | 9 |
| IV | 1.00 | 0.50 | 9 |
| **V** | 1.00 | 0.50 | 3 |
| all | 0.997 | 0.498 | 9 |

Notes: $r^2$, allele frequency correlations.

Classified as a relict endangered species in China [9], *T. sinense* has been the focus of recent conservation efforts through nature reserve establishment to safeguard its germplasm resources. Maintaining intact genetic diversity is critical for preserving evolutionary potential to adapt to future environmental changes. Conservation strategies include: (1) integrated in situ/ex situ preservation and reintroduction for low-diversity populations (HBY, HG, BSJ, BH); (2) germplasm nurseries with seed collection for high-diversity populations (SHS, BDGS, FP, WL); (3) gene flow augmentation from large source to small sink populations to mitigate inbreeding depression and genetic drift [58,59]; (4) clade-specific management units aligned with the five evolutionary clusters identified via admixture and DAPC analyses; and (5) genetic rescue interventions (immigration, pollen exchange, controlled hybridization) for isolated populations. *Ex situ* preservation and reintroduction are particularly vital for sustaining genetic diversity in this endangered species.

## Conclusions

The current study examined the population genetic structures and conservation strategies for the relict and endangered plant, *T. sinense*, in central and southern China. Using genome-wide SNPs (RAD-seq) analysis, we found low genetic diversity and high differentiation in *T. sinense*. Additionally, we confirmed the presence of five conservation management units, with the Hengduan, Wuling, Minshan, and Qinling-Daba Mountains serving as refuges in China. The populations in the Qinling-Daba Mountains are considered the original center of *T. sinense*. This study represents the first exploration of genetic information through SNP analysis. Furthermore, we provide theoretical guidance for future conservation efforts.

## Supporting information

**S1 Fig. The quality distribution of sequencing.**
(TIF)

**S2 Fig. Clades from 1–27 based on Admixture analysis.**
(PDF)

**S1 Table. Detail informations of *T. sinense* samples.**
(XLSX)

**S2 Table. The raw data of the 164 *T. sinense* individuals.**
(XLSX)

**S3 Table. Single locus nucleotide diversity (π) value.**
(XLSX)

**S4 Table. 164 *T. sinense* individuals based on the adegenet analysis.**
(XLSX)

**S5 Table. K value and CV error in Admixture analysis.**
(XLSX)

**S6 Table. Admixture determined clades of 164 *T. sinense* individuals.**
(XLSX)

**S7 Table. Principal components (PC1, PC2 and PC3) of 164 *T. sinense* individuals.**
(XLSX)

## Acknowledgments

The authors are also grateful to 25 nature reserves and plant protection stations for their help with the geographic information of the distribution of *T. sinense*. This study has been supported by National Natural Science Foundation of China (32070371 to X.G., 31760127 to Q.L), High-level graduate research project, Xizang University (2021-GSP-B019 to Z.T.), and authors thank them here. All claims expressed in this article are solely those of the author and do not necessarily represent those of their affiliated organizations, or those of the publisher, the editors, and the reviewers. Any product that may be evaluated in this article, or claim that may be made by its manufacturer, is not guaranteed or endorsed by the publisher.

## Author contributions

**Formal analysis:** Zhong–Qiong Tian.

**Funding acquisition:** Zhong–Qiong Tian, Qiong La, Xiaohong Gan.

**Investigation:** Chao–Yang Jiang, Yu–Min Shu, Huan Zhang, Qiong La, Xiaohong Gan.

**Project administration:** Qiong La, Xiaohong Gan.

**Resources:** Zhong–Qiong Tian.

**Software:** Zhong–Qiong Tian.

**Supervision:** Xiaohong Gan.

**Validation:** Chao–Yang Jiang, Yu–Min Shu, Huan Zhang, Qiong La.

**Visualization:** Zhong–Qiong Tian, Xiaohong Gan.

**Writing – original draft:** Zhong–Qiong Tian.

**Writing – review & editing:** Qiong La, Xiaohong Gan.

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
