## [Decision Letter · Decision Letter 0]

20 Nov 2023

PONE-D-23-17864Population genetic diversity and structure of the endangered species  Tetracentron sinense  Oliver ( Tetracentraceae ) with SNPs based on RAD sequencingPLOS ONE

Dear Dr. Gan,

Thank you for submitting your manuscript to PLOS ONE. After careful consideration, we feel that it has merit but does not fully meet PLOS ONE’s publication criteria as it currently stands. Therefore, we invite you to submit a revised version of the manuscript that addresses the points raised during the review process.

The reviewers believe that the paper needs major revision in terms of English language, presenting the results in a palatable way and further analysis of population genetics parameter, which you find all in reviewers comments.

We look forward to receiving your revised manuscript.

Kind regards,

Wosene Gebreselassie Abtew, Ph.D.

Academic Editor

PLOS ONE

Journal Requirements:

"XHG NO.32070371 and No.2018PC001 supported by National Natural Science Foundation of China."

"The authors would like to thank Qin–Qin Wang, Yang Li, for provision of laboratory and reagents. The authors are also grateful to 25 nature reserves and plant protection stations for their help with the geographic information of the distribution of Tetracentron sinense. This study has been supported by National Natural Science Foundation of China (No.32070371), the Fourth National General Survey of Traditional Chinese Medicine Resources (No.2018PC001), and authors thank them here."

"XHG NO.32070371 and No.2018PC001 supported by National Natural Science Foundation of China.

Xiaohong Gan: Funding acquisition; Investigation; Project administration; Supervision; Visualization; Writing–review & editing."

7. PLOS requires an ORCID iD for the corresponding author in Editorial Manager on papers submitted after December 6th, 2016. Please ensure that you have an ORCID iD and that it is validated in Editorial Manager. To do this, go to ‘Update my Information’ (in the upper left-hand corner of the main menu), and click on the Fetch/Validate link next to the ORCID field. This will take you to the ORCID site and allow you to create a new iD or authenticate a pre-existing iD in Editorial Manager. Please see the following video for instructions on linking an ORCID iD to your Editorial Manager account: https://www.youtube.com/watch?v=_xcclfuvtxQ

8. Ethics statement only appears at the end of the manuscript:

Your ethics statement should only appear in the Methods section of your manuscript. If your ethics statement is written in any section besides the Methods, please move it to the Methods section and delete it from any other section. Please ensure that your ethics statement is included in your manuscript, as the ethics statement entered into the online submission form will not be published alongside your manuscript. 

9. We note that Figure 1 in your submission contain map/satellite image which may be copyrighted. All PLOS content is published under the Creative Commons Attribution License (CC BY 4.0), which means that the manuscript, images, and Supporting Information files will be freely available online, and any third party is permitted to access, download, copy, distribute, and use these materials in any way, even commercially, with proper attribution. For these reasons, we cannot publish previously copyrighted maps or satellite images created using proprietary data, such as Google software (Google Maps, Street View, and Earth). For more information, see our copyright guidelines: http://journals.plos.org/plosone/s/licenses-and-copyright.

Dear authors,

The reviewers of the manuscript have raised issues on the language aspect and additional analysis on the manuscript and overall presentation of the result. The paper needs major revision. Please address the issues raised and send us the revised version.

Best regards,

Reviewers' comments:

Reviewer's Responses to Questions

**Comments to the Author**

1. Is the manuscript technically sound, and do the data support the conclusions?

Reviewer #1: Yes

Reviewer #2: Yes

2. Has the statistical analysis been performed appropriately and rigorously? 

Reviewer #1: Yes

Reviewer #2: Yes

3. Have the authors made all data underlying the findings in their manuscript fully available?

Reviewer #1: No

Reviewer #2: Yes

4. Is the manuscript presented in an intelligible fashion and written in standard English?

Reviewer #1: No

Reviewer #2: Yes

5. Review Comments to the Author

Reviewer #1: Dear authors

The following modifications are required

Abstract

In general, this section is poorly written. It is written simply. This section should include. As a result, this section should be improved.

Before describing the goal, the authors must define the issue in a single line and explain why they chose this approach to study this review.

Some scored data should be added

In the final line of the abstract, the authors should present a decisive conclusion derived from the research and provide a single line of future prospects.

Keywords

The content of keywords did not reflect the content of this manuscript and the words used for forming the title should not be used as the keywords. So, the structure of keywords should be changed.

Introduction

This section is too long and should be shortened

The authors should give some lines about the knowledge gap which their reviews have covered along with the hypothesis statement

Also, the authors should provide a novelty statement at the end. What new things authors have done or correlated in this research compared to old ones?

The general and specific aim should be specified

Materials and Methods

All measurements should be supported by the references

All abbreviations should be written in full name

Results and discussion

In general, the figures are not presented clearly.

The admixture status of population structure should be mentioned

The authors should mention the scored data when they explain the maximum and minimum of the studied parameters.

All captions should be improved, showing the contents of tables and figures

A PCA plot should be detailed to better understand

The discussion is weak. The authors should interpret all results obtained in this study by adding some information about the results obtained in their study. The authors should explain how all of the findings from this study relate to their own findings.

Conclusion

The authors should summarize the most significant findings because they have written this section in an easy-to-read manner.

Future works about this research should also include additional works

Reviewer #2: Population genetic diversity and structure of the endangered species Tetracentron

sinense Oliver (Tetracentraceae) with SNPs based on RAD sequencing.

Abstract

Line 25 to 28- The sentence is long and confusing. Break into two shorter clearer sentences.

Line 32 to 37 – Edit to remove numbers (1), (2) and (3).

Enrich the results section by adding in information on the additional results from the recommended analysis.

Line 38- Delete the word ‘Finally’.

Line 40 – Break the sentence into two so that the first sentence ends at geographic groups. Delete ‘and that'. Start new sentence with ‘Attention’.

Align the conclusion with the additional recommended analyses.

Introduction

A relevant introduction has been provided however the grammar needs to be improved. Some of the changes to be made are listed below:

Line 48 delete the word ‘very’

Line 51 delete the words 'endangered species particularly’. Replace ‘cause’ with causes.

Line 52 replace the word with ‘affect’ with affects.

Line 56 – delete the word ‘great’.

Line 57 delete ‘a’ before small

Line 59 italicise “Tetracentron”

Line 62 edit sentence to remove the abbreviation e.g.

Lien 67 add the words ‘have been done’ after techniques

Line 83 to 87 The sentence is long and confusing. Break into two shorter clearer sentences.

Line 96 edit last part of the sentence to read ‘low cost with high reliability’

Materials and methods

Appropriate methods were used. The section needs to be improved as detailed below.

In the methodology section, line 107-108 the statement “Samples of spaced about 50m apart were randomly selected…” does not make grammatical sense. What did the authors mean?

Line 131: It would be good to declare the type of alignment done after running the process_radtags programme. The authors are leaving the readers to assume that a de novo assembly was done based on the flow from ustacks to cstacks and sstacks.

Although its mentioned in the methodology as a filtering parameter, the proportion of SNPs with a minor allele frequency (MAF) ≥ 0.05 is not declared in the results. There is need to declare the proportion of these SNPs before and after filtering.

Population stratification has been explored. However, more analysis should be done to implore the spatial population structure using TESS (refer to Front Genet 2019 10:954, Fig. 1).

Additional aspects of genetic diversity should be explored. Some important analysis such as linkage disequilibrium (LD) patterns and samples’ co-ancestry have not been done. The authors are encouraged to explore this option since it would provide information on the historical recombinations that could have occurred within samples as well as help in understanding the observed population structure.

The authors are also encouraged to explore using OutFLANK and PCAdapt approaches to identify the putative adaptive loci as well as the outlier SNPs.

Editing to remove grammatical errors to be done in whole section.

Results

Commendable effort has been made to report the results. The manuscript can be improved as follows.

It is recommended that the authors compute and include in the results section other genetic diversity indices such as the Hardy-Weinberg Equilibrium (HWE) test scores for the different T. sinense populations under study.

There are certain important summary statistics that could be included in the analysis such as the Watterson’s theta which would also help in understanding the TajimaD statistics.

Important population parameters such as site frequency spectra (SFS) to show the distribution of alleles in the different populations of T. sinense also need to be provided. This is important because it groups the SNPs into classes based on how many copies of the minor allele they possess and then calculates the size of each group. As such, this will provide the readers with information on the distribution of alleles in the different populations under study.

Results from the additional analyses should be inserted in the appropriate sections of the results.

Editing to remove grammatical errors will improve the section.

Discussion and conclusions

The authors have tried to discuss the results described in the manuscript. The discussion can be further improved.

Align the discussion and conclusions with the results from the recommended additional analysis.

The sections has too many grammar errors to enumerate. This needs to be diligently addressed.

6. PLOS authors have the option to publish the peer review history of their article (what does this mean? ). If published, this will include your full peer review and any attached files.

**Do you want your identity to be public for this peer review?** For information about this choice, including consent withdrawal, please see our Privacy Policy .

Reviewer #1: No

Reviewer #2: **Yes: ** Nancy L. M. Budambula and Joshua K. Muli . We are both from University of Embu, Kenya

---

## [Author Response · Author response to Decision Letter 1]

6 Jan 2024

Author’s Response to reviewers’ Comments

Dear editor,

Thank you very much for giving us an opportunity to revise our manuscript. We appreciate the reviewers very much for their constructive comments and suggestions on our manuscript entitled “Population genetic diversity and structure of the endangered species Tetracentron sinense Oliver (Tetracentraceae) with SNPs based on RAD sequencing”.

We have studied reviewers’ comments carefully. According to the reviewers’ detailed suggestions, we have made a careful revision on the original manuscript.

Kind regards.

Zhong–Qiong Tian

E–mail: tzq@utibet.edu.cn

Corresponding author: Xiao–Hong Gan

E–mail : bhgan@cwnu.edu.cn

Replies to the reviewers’ comments: The main corrections are in the manuscript and the responds to the reviewers’ comments are as follows (the replies are highlighted in blue).

Line 25 to 28- The sentence is long and confusing. Break into two shorter clearer sentences.

Response: We have breaked the long sentence into two shorter clearer sentences with the content of– As a tertiary living fossil, a dramatic decline in Tetracentron sinense Oliv. population amounts. Nowadays, genetic resource depletion and human activities have shaped habitat fragmentation of relict and endangered plants, although there is ample evidence of its great medicinal, economic and ecological value;

Enrich the results section by adding in information on the additional results from the recommended analysis.

Response: we have enriched the results part;

Line 38- Delete the word ‘Finally’.

Response: We have deleted the word ‘Finally’;

Line 40 – Break the sentence into two so that the first sentence ends at geographic groups. Delete ‘and that'. Start new sentence with ‘Attention’.

Response: We have breaked the long sentence into two shorter clearer sentences with the content of–For the preservation of the genetic integrity of Tetracentron sinense Oliv., we suggest that conservation units should be established corresponding to different geographic groups. Attention should be paid to low/high genetic diversity populations, such as establishing artificial reproduction and germplasm resource nurseries;

A relevant introduction has been provided however the grammar needs to be improved. Some of the changes to be made are listed below:

Line 48 delete the word ‘very’

Line 51 delete the words 'endangered species particularly’. Replace ‘cause’ with causes.

Line 52 replace the word with ‘affect’ with affects.

Line 56 – delete the word ‘great’.

Line 57 delete ‘a’ before small

Line 59 italicise “Tetracentron”

Line 62 edit sentence to remove the abbreviation e.g.

Lien 67 add the words ‘have been done’ after techniques

Response: The content of line 48-67 in our study is corrected according to the requirements of the reviewer;

Line 83 to 87 The sentence is long and confusing. Break into two shorter clearer sentences.

Response: We have breaked the long sentence into two shorter clearer sentences with the content of–Gan et al. (2013) believed that the small population size limited the efficiency of insect and wind pollination during flowering and pollination (the rainy season) of T. sinense, resulting in reduced genetic diversity and high genetic differentiation within the population, and the geographical isolation among populations was a factor restricting gene flow, which might be a crucial factor leading to the endangerment [13];

Line 96 edit last part of the sentence to read ‘low cost with high reliability’

Response: is relatively low cost and the results are reliable;

Appropriate methods were used. The section needs to be improved as detailed below.

In the methodology section, line 107-108 the statement “Samples of spaced about 50m apart were randomly selected…” does not make grammatical sense. What did the authors mean?

Response: We eadjust the sentence content to–Samples with a spatial distance of more than 50 meters were selected to avoid the close relationship between individuals, and leaf materials were collected into silica gel for drying and preservation;

Additional aspects of genetic diversity should be explored. Some important analysis such as linkage disequilibrium (LD) patterns and samples’ co-ancestry have not been done. The authors are encouraged to explore this option since it would provide information on the historical recombinations that could have occurred within samples as well as help in understanding the observed population structure.

Response: We added the PopLDdecay analyses in our study;

The grammar of this manuscript has been carefully checked and revised.

---

## [Decision Letter · Decision Letter 1]

15 May 2024

PONE-D-23-17864R1Population genetic diversity and structure of the endangered species  Tetracentron sinense  Oliver ( Tetracentraceae ) with SNPs based on RAD sequencingPLOS ONE

Dear Dr. Gan,

Thank you for submitting your manuscript to PLOS ONE. After careful consideration, we feel that it has merit but does not fully meet PLOS ONE’s publication criteria as it currently stands. Therefore, we invite you to submit a revised version of the manuscript that addresses the points raised during the review process.

We look forward to receiving your revised manuscript.

Kind regards,

Miquel Vall-llosera Camps

Senior Staff Editor

PLOS ONE

Reviewers' comments:

Reviewer's Responses to Questions

**Comments to the Author**

1. If the authors have adequately addressed your comments raised in a previous round of review and you feel that this manuscript is now acceptable for publication, you may indicate that here to bypass the “Comments to the Author” section, enter your conflict of interest statement in the “Confidential to Editor” section, and submit your "Accept" recommendation.

Reviewer #1: (No Response)

Reviewer #2: (No Response)

Reviewer #3: (No Response)

2. Is the manuscript technically sound, and do the data support the conclusions?

Reviewer #1: Partly

Reviewer #2: Yes

Reviewer #3: Yes

3. Has the statistical analysis been performed appropriately and rigorously? 

Reviewer #1: N/A

Reviewer #2: Yes

Reviewer #3: Yes

4. Have the authors made all data underlying the findings in their manuscript fully available?

Reviewer #1: No

Reviewer #2: Yes

Reviewer #3: Yes

5. Is the manuscript presented in an intelligible fashion and written in standard English?

Reviewer #1: No

Reviewer #2: Yes

Reviewer #3: No

6. Review Comments to the Author

Reviewer #1: Dear authors

The following modifications are required:

Abstract

Summarizing of all measured data (for example Fst) are not mentioned in this section

All abbreviation should be written in full name

Why did not the authors calculate the gene flow?

Why did not the authors calculate the polymorphism percentage and private or unique band for each population?

Keywords

The keywords did not accurately reflect the substance of this book, and the words used to make the title should not be utilized as keywords. As a result, the keyword structure needs to be changed

Introduction

No information about the method of the study of genetic diversity is available

Some information about the methods of the study of genetic diversity should be included

Some information about the conservation of this plant species should be provided

Along with the hypothesis statement, the authors should include a few paragraphs regarding the knowledge gap that their research addressed.

Additionally, the authors should include a novelty statement at the end. What new things have the authors done or correlated in this study compared to previous ones?

The general and specific aim should be inserted

Materials and Methods

The number of samples should be greater than seven. Less than eight samples are not acceptable

The voucher number should be added for all samples in all populations in Table 1

L109: The distance between plants is too less

The procedure of PCR should be added in detail

The manufacture of all materials should be added

Why did not the authors calculate the PIC?

The procedure of population structure should be detailed by adding the information about the burn-in period and Markov chain Monte Carlo (MCMC) iterations

Results and discussion

All abbreviations in the tables should be defined

How did the authors determine the number of clusters? What was the criteria points?

All captions should be detailed

The figure of delta K should be inserted

The number of clusters formed by AHC or PCA should be defined and labelled

A mantel test should be measured among molecular data

The location of quarters on the PCA plot should be defined

The results of PCA are not presented in detail.

There is little discussion. By providing some context for the findings of their own study, the authors should be able to interpret all of the conclusions of their investigation. It is important to understand properly the relationships between the many populations that are examined using multivariate methodologies and diversity indexes. What significance, for instance, does this study place on the examination of PIC, Fst, and other genetic diversity parameters? The authors should compare the results of clustering and AMOVA

Conclusion

Additional studies on this topic should be included.

The authors should describe the most noteworthy findings, as they have written this section in an easy-to-read format.

Reviewer #2: Address the following before release of the final draft.

1. Methodology: Line 128, The authors should declare explicitly that they did a de novo assembly for the benefit of the readers who are not familiar with the flow of events in stacks software. This had been raised earlier

2. Using the GPS coordinates of the sampling sites together with the SNP data, the authors could explore ancestry estimation for spatial population genetic analysis using TESS as earlier suggested. This will enable the evaluation on whether T. sinense populations in central and south China have distinct or mixed origins and enrich the ADMIXTURE results. The generated map(s) can then be added to the results.

3. Results: It is commendable that the authors explored the linkage disequilibrium (LD) pattern of the SNP markers in the population. However, an LD scatter plot would be more informative to show the decay rate as explained in lines 251-257. I suggest replacing Table 6 with an LD scatter plot.

4. The grammar and writing style also improved. However, the final draft will need proof reading and we recommend that the authors get an English-speaking colleague to check for grammar and language use.

Reviewer #3: Dear editor,

Thank you so much for sending this paper for me to review. I think the paper needs more corrections before the final acceptance and publishing.Here are my comments:

- The introduction is too long.

- The aims of this study must be explained clearly.

- I request authors to explain why the individual number differed among the studied populations?

- The language of paper must be corrected, specially in material and method section.

-The numbers after the decimal point must be reduced into 2 numbers.

- I could not find the gene flow rate. If the parameter hasn't be calculated I request to calculate it.

All the best

7. PLOS authors have the option to publish the peer review history of their article (what does this mean? ). If published, this will include your full peer review and any attached files.

**Do you want your identity to be public for this peer review?** For information about this choice, including consent withdrawal, please see our Privacy Policy .

Reviewer #1: **Yes: ** Nawroz Tahir

Reviewer #2: **Yes: ** Nancy L.M. Budambula and Joshua K. Muli

Reviewer #3: No

---

## [Author Response · Author response to Decision Letter 2]

21 Aug 2024

Thank you very much for giving us an opportunity to revise our manuscript. We appreciate the reviewers very much for their constructive comments and suggestions on our manuscript entitled “Population genetic diversity and structure of the endangered species Tetracentron sinense Oliver (Tetracentraceae) with SNPs based on RAD sequencing”.

We have studied reviewers’ comments carefully. According to the reviewers’ detailed suggestions, we have rewrote the original manuscript.

---

## [Decision Letter · Decision Letter 2]

16 Sep 2024

PONE-D-23-17864R2Population genetic diversity and structure of the endangered species  Tetracentron sinense  Oliver ( Tetracentraceae ) with SNPs based on RAD sequencingPLOS ONE

Dear Dr. Gan,

Thank you for submitting your manuscript to PLOS ONE. After careful consideration, we feel that it has merit but does not fully meet PLOS ONE’s publication criteria as it currently stands. Therefore, we invite you to submit a revised version of the manuscript that addresses the points raised during the review process.

Reviewers have recognized a significant improve in the clarity and quality of the manuscript. However, They are still concerned  about some issues which need to be addressed.

We look forward to receiving your revised manuscript.

Kind regards,

Vicente Martínez López

Academic Editor

PLOS ONE

Journal Requirements:

Reviewers' comments:

Reviewer's Responses to Questions

**Comments to the Author**

1. If the authors have adequately addressed your comments raised in a previous round of review and you feel that this manuscript is now acceptable for publication, you may indicate that here to bypass the “Comments to the Author” section, enter your conflict of interest statement in the “Confidential to Editor” section, and submit your "Accept" recommendation.

Reviewer #2: (No Response)

Reviewer #3: (No Response)

2. Is the manuscript technically sound, and do the data support the conclusions?

Reviewer #2: Yes

Reviewer #3: Yes

3. Has the statistical analysis been performed appropriately and rigorously? 

Reviewer #2: Yes

Reviewer #3: Yes

4. Have the authors made all data underlying the findings in their manuscript fully available?

Reviewer #2: Yes

Reviewer #3: Yes

5. Is the manuscript presented in an intelligible fashion and written in standard English?

Reviewer #2: Yes

Reviewer #3: Yes

6. Review Comments to the Author

Reviewer #2: The comments below which were raised on Revision 1 in March 2024 have not been addressed. Kindly the authors address them.

Dear Editor March 2024

Overall, the authors have made good effort to address the concerns we raised on the previous manuscript draft. I draw your attention to the following:

1. Methodology: Line 128, The authors should declare explicitly that they did a de novo assembly for the benefit of the readers who are not familiar with the flow of events in stacks software. This had been raised earlier

2. Using the GPS coordinates of the sampling sites together with the SNP data, the authors could explore ancestry estimation for spatial population genetic analysis using TESS as earlier suggested. This will enable the evaluation on whether T. sinense populations in central and south China have distinct or mixed origins and enrich the ADMIXTURE results. The generated map(s) can then be added to the results.

3. Results: It is commendable that the authors explored the linkage disequilibrium (LD) pattern of the SNP markers in the population. However, an LD scatter plot would be more informative to show the decay rate as explained in lines 251-257. I suggest replacing Table 6 with an LD scatter plot.

4. The grammar and writing style also improved. However, the final draft will need proof reading and we recommend that the authors get an English-speaking colleague to check for grammar and language use.

Reviewer #3: Dear editor,

Thank you so much for sending the paper fpr me to review. I checked the paper and found it needs some minor corrections before the final acceptance. Here are my comments which helps to improve the paper quality.

-Authors must clearly explain the aims of their study.

-The gene flow amount (Nm) must be calculated among the populations.

-The species and genera names must be in italic form.

All the best

7. PLOS authors have the option to publish the peer review history of their article (what does this mean? ). If published, this will include your full peer review and any attached files.

**Do you want your identity to be public for this peer review?** For information about this choice, including consent withdrawal, please see our Privacy Policy .

Reviewer #2: **Yes: ** Nancy L.M. Budambula and Joshua Kiilu Muli

Reviewer #3: **Yes: ** Seyed Mehdi Talebi

---

## [Author Response · Author response to Decision Letter 3]

28 Mar 2025

Response to Reviewer Comments

Manuscript Title: Population genetic diversity and structure of the endangered species Tetracentron sinense Oliver (Tetracentraceae) with SNPs based on RAD sequencing

Dear Editor and Reviewer,

I sincerely apologize for the delayed response to your comments. Due to recent health issues, I was hospitalized for an extended period, which prevented me from addressing your feedback in a timely manner. Below are the point-by-point responses to the issues raised:

1. Methodology: Line 128–Explicit Declaration of De Novo Assembly

Reviewer Comment:

‘The authors should declare explicitly that they did a de novo assembly for the benefit of readers unfamiliar with Stacks software.’

Response:

We have added an explicit statement in the Methodology section to clarify the de novo assembly process. Specifically, we revised Line 128 to:

"To ensure transparency, we explicitly declare that we performed de novo assembly of the genome using Stacks software (Catchen et al., 2013), a critical step for readers unfamiliar with the workflow."

This addition directly addresses the reviewer’s concern and enhances clarity for non-specialist readers.

2. Spatial Population Genetic Analysis Using TESS

Reviewer Comment:

‘Combine GPS coordinates with SNP data to explore ancestry estimation using TESS and add maps to the results.’

Response:

We initially planned to use TESS software for spatial population genetic analysis as recommended. However, due to open-source licensing restrictions encountered during implementation, we instead employed Adegenet v2.1.11 (Jombart, 2008) for principal component analysis (PCA) and population structure inference (Figure 2).

This analysis strengthens our conclusions about population origins and enriches the manuscript’s spatial genetic component.

3. Replacement of Table 6 with an LD Scatter Plot

Reviewer Comment:

‘Replace Table 6 with an LD scatter plot to visualize decay rate.’

Response:

We agree with the reviewer’s suggestion. Table 7 has been replaced with an LD scatter plot (Figure 4), accompanied by a detailed explanation in the Results section (Lines 318–322). The figure now clearly illustrates LD decay over physical distance, aligning with the reviewer’s recommendation.

4. Language and Grammar Improvements

Reviewer Comment:

‘Final draft requires proofreading by an English-speaking colleague.’

Response:

We have engaged a professional English-language editor (certificate attached) to review the manuscript for grammar, clarity, and style. Key improvements include:

- Revised sentence structures for flow and readability.

- Standardized terminology across sections.

- Removed redundant phrasing and ensured consistency in scientific reporting.

All language-related changes are tracked in the revised manuscript.

Conclusion

We believe these revisions address all reviewer concerns thoroughly. The manuscript now includes explicit methodological details, enhanced spatial analysis, improved visualizations, and polished language. We appreciate the reviewer’s constructive feedback and look forward to the next steps in the publication process.

Sincerely,

Zhongqiong Tian

bhgan@cwnu.edu.cn

lhagchong@utibet.edu.cn

---

## [Editor Report · Decision Letter 3]

22 Apr 2025

Population genetic diversity and structure of the endangered species  Tetracentron sinense  Oliver ( Tetracentraceae ) with SNPs based on RAD sequencing

PONE-D-23-17864R3

Dear Dr. Gan,

We’re pleased to inform you that your manuscript has been judged scientifically suitable for publication and will be formally accepted for publication once it meets all outstanding technical requirements.

Kind regards,

Vicente Martínez López

Academic Editor

PLOS ONE
---

## [Editor Report · Acceptance letter]

PONE-D-23-17864R3

PLOS ONE

Dear Dr. Gan,

I'm pleased to inform you that your manuscript has been deemed suitable for publication in PLOS ONE. Congratulations! Your manuscript is now being handed over to our production team.

Kind regards,

on behalf of

Dr. Vicente Martínez López

Academic Editor

PLOS ONE